behaviour, ecology

collective behaviour, recruitment mechanisms, self-organization, polydomy

**Author for correspondence:**
Elva J. H. Robinson
e-mail: elva.robinson@york.ac.uk

# From foraging trails to transport networks: how the quality-distance trade-off shapes network structure

Valentin Lecheval[1], Hannah Larson[2], Dominic D. R. Burns[1], Samuel Ellis[3], Scott Powell[4], Matina C. Donaldson-Matasci[2] and Elva J. H. Robinson[1]

[1]Department of Biology, University of York, UK
[2]Department of Biology, Harvey Mudd College, Claremont, CA, USA
[3]Centre for Research in Animal Behaviour, University of Exeter, UK
[4]Department of Biological Sciences, George Washington University, Washington, DC, USA

VL, 0000-0001-6041-2718; DDRB, 0000-0003-2299-2453; SE, 0000-0001-9019-6040;
SP, 0000-0001-5970-8941; MCD-D, 0000-0002-9877-384X; EJHR, 0000-0003-4914-9327

Biological systems are typically dependent on transportation networks for the efficient distribution of resources and information. Revealing the decentralized mechanisms underlying the generative process of these networks is key in our global understanding of their functions and is of interest to design, manage and improve human transport systems. Ants are a particularly interesting taxon to address these issues because some species build multi-sink multi-source transport networks analogous to human ones. Here, by combining empirical field data and modelling at several scales of description, we show that pre-existing mechanisms of recruitment with positive feedback involved in foraging can account for the structure of complex ant transport networks. Specifically, we find that emergent group-level properties of these empirical networks, such as robustness, efficiency and cost, can arise from models built on simple individual-level behaviour addressing a quality-distance trade-off by the means of pheromone trails. Our work represents a first step in developing a theory for the generation of effective multi-source multi-sink transport networks based on combining exploration and positive reinforcement of best sources.

## 1. Introduction

Transport networks are encountered across all biological levels, from the cardiovascular and respiratory systems in vertebrates [1], to multicellular fungi [2] and social insect nests [3]. Biological transport networks achieve global optimization without centralized control, through self-organization [4,5]. Studying the cost, robustness and efficiency of biological transport networks can therefore aid in designing, managing and improving human systems that transport people, energy, products or information.

Multi-source multi-sink transport networks, in which commodities and passengers flow within complex networks between multiple locations, pose particular challenges for design and management (e.g. [6]). These human transport systems could benefit from a better understanding of the mechanisms underlying the emergence and dynamics of biological multi-source multi-sink transport networks, shaped by evolutionary processes. Impressive transport networks of this type are formed by colonies of certain 'polydomous' ant species, which are distributed across multiple spatially separated nests connected to each other and to foraging patches by a network of trails [7]. Several polydomous species form nest networks with global properties making the structure efficient, low cost in terms of total trail distance yet still robust to some trail disruption [8,9]. At the colony level, polydomous ants

face the challenge of distributing resources from multiple sources to the many nests of the network, resonating with human multi-source multi-sink problems.

Ecologically dominant wood ants (*Formica rufa* group) can form especially stable and empirically tractable resource-exchange networks that connect numerous nests and food patches [10,11]. Workers forage from their habitual nest of origin either to trees (food patches providing hemipteran honeydew), or to other nests of the colony that they treat as food sources [12–15]. Hence, the resulting multi-source multi-sink transport networks are hypothesized to result from a self-organized process occurring at the scale of the nests [9,16,17]. However, little is known about the possible mechanisms underlying a self-organized process leading to such networks. Pre-existing ant foraging mechanisms are hypothesized to facilitate the evolution of polydomous resource-exchange networks [15], but whether there is a link between these mechanisms and network structure is uncertain.

In social insects, foraging is often a complex decision-making process consisting of retrieving resources from an uncertain environment to a central depot—the nest. At the macroscopic scale, colonies face the challenge of optimally allocating workers to food resources to maximize resource intake while minimizing transportation costs [18]. An important part of this challenge lies in the quality-distance trade-off, whereby colonies may benefit from allocating workers to either the most valuable resources or the closest ones (reducing worker loss and time/energetic expenditure) [19]. Recruitment with positive feedback in foraging ants usually favours the highest quality resources [20–24] and/or the closest resources [25,26]. Theoretical models combined with robot experiments show that pheromone laying is a reliable behavioural mechanism to optimally address quality-distance trade-offs [27,28].

Here, we aim to characterize the candidate mechanisms leading to the formation of effective multi-source multi-sink transport networks. Specifically, we investigate the hypothesis that well-studied foraging mechanisms could explain the structure of resource-exchange networks in polydomous ants. We use field data on wood ants to (i) develop a pheromone-recruitment model whose predictions are evaluated in terms of choices regarding food sources of varying distance and quality at the level of the colony. These predictions are then used in a second model of network morphogenesis at the colony level investigating (ii) to what extent recruitment mechanisms with positive feedback can account for the structural properties of polydomous resource-exchange networks. Rather than developing a generative network model optimizing macroscopic metrics, we instead test the hypothesis that, through self-organization and decentralized decision-making, simple and local rules of recruitment can give rise to complex multi-source multi-sink networks balancing cost, efficiency and robustness at the colony level. The methodology we follow, combining modelling at several scales of description and empirical field data, is summarized in the electronic supplementary material, figure S1.

## 2. Pheromone-recruitment model (methods and results)

The combined effects of food quality and distance on foraging traffic in complex and dynamic environments have not been thoroughly investigated. In particular, we do not know how the number of ants on each trail (i.e. the allocation of workers at the colony level) varies as a function of the quality of the food source and the length of the trail. The study of this variation is based on the idea that the presence of an ant on a trail can be seen at the colony level as a vote of the individual in favour of the resource connected to the trail. The higher this number, the more the colony allocates workers to a food source and votes in its favour. The variation in the number of ants committed to a trail changes with the quality of sources depending on three mechanisms: spontaneous discovery of food sources owing to scouting, recruitment with positive feedback, and ants leaving the trail [29–31]. In this model (referred to as the Sumpter and Beekman model thereafter), the change in the number of ants $X_i$ committed to a trail $i$ in a situation where the focal nest is connected to $J$ food patches of quality $q$ is

$$\frac{dX_i}{dt} = [A + B(q_i, X_i)]\left(N - \sum_{j=1}^{J} X_j\right) - S(X_i)X_i, \qquad (2.1)$$

where $A$, $B$ and $S$ are the per capita rate of, respectively, spontaneous discoveries of food sources, recruitment to food source $i$ and leaving the trail $i$ [29,30]. The term $\left(N - \sum_{j=1}^{J} X_j\right)$, with $N$ the number of ants available for foraging in the nest, represents the number of ants not committed to any trail. However, in the Sumpter and Beekman model, the three mechanisms do not depend on distance, even though distance probably has a great influence on foraging success.

To examine how the quality of a food source and its distance from the nest interact to influence ant foraging, we reformulate the Sumpter and Beekman model to make all three mechanisms dependent not just on quality, but also distance. The model can be reformulated as

$$\frac{dX_i}{dt} = [A(d_i) + B_i(d_i, q_i, X_i)]\left(N - \sum_{j=1}^{J} X_j\right) - S(d_i, q_i, X_i)X_i, \quad (2.2)$$

where $d_i$ and $q_i$ are the distance to and the quality of the source $i$.

How does the per capita rate of spontaneous discovery of food sources $A(d_i)$ depend on distance? First, we assume that workers from a given nest looking for food resources (so-called scouts) are not homogeneously distributed in space: they are more likely to be found closer to their nest of origin than elsewhere. We checked this important assumption in the electronic supplementary material, section D by analysing available published data of single *Temnothorax albipennis* ants exploring the area around their nest entrance [32]. A possible underlying behavioural mechanism is that scouts that are still looking for food have a probability of returning to their home nest [33] that remains constant over time (see discussion in the electronic supplementary material, section D). With such a behaviour, the distribution of the distance of scouting ants from their home nest will be exponential. While some ant species may use chemical cues to direct exploration [34], for simplicity, we omit the effects of such behaviour. Thus, we write the rate of spontaneous discovery of food sources:

$$A(d_i) = \alpha e^{-\gamma_1 d_i}, \qquad (2.3)$$

with $\alpha$ controlling the per capita rate of spontaneous discoveries and $\gamma_1$ a coefficient of inverse length dimension reflecting the range of foraging of scouts.

We also assume that recruitment mechanisms mainly rely on pheromone trails, at least in the emergence phase of the trail; other processes may be involved after the trail formation, such as workers remembering the location of the food source. When scouts find a food source, they lay pheromones on their way back to the nest, in order to recruit other workers. This mechanism can eventually lead to the formation of a trail, through positive feedback when reinforced by many ant passages. The trail may not be reinforced if the rate of ants returning to the nest via the emerging trail is too low because of the combined effects of diffusion and evaporation of pheromones. Increasing the distance between a food source and the nest will make the reinforcement of the trail harder owing to the evaporation of pheromones: for the trail to be equally attractive to recruit other workers when the distance is increased, more ants returning and depositing pheromones are required [26]. In this condition, per capita rate of recruitment is inversely proportional to distance. Thus, we set the per capita rate of recruitment to food source $i$ for uncommitted ants to

$$B_i(d_i, q_i, X_i) = \frac{\gamma_2}{d_i} \beta_i X_i, \tag{2.4}$$

where $\gamma_2$ is the range of the recruitment activity (length dimension), reflecting the range over which ants sense pheromones, and $\beta_i = \eta q_i$ indicates how much each ant along the trail to $i$ contributes to the rate of recruitment to source $i$, which is proportional to quality $q_i$.

As for the per capita rate of ants leaving the trail, we assume two distinct effects: (i) there is a probability for ants to leave a trail they are following, which is constant per distance unit; and (ii) this probability decreases when the strength of the pheromone trail $(\gamma_3/d_i)\beta'_i X_i$ increases. $\beta'_i = \eta' q_i$ depicts the proportional relationship between the pheromone strength of the trail and quality and $\gamma_3$ is the range of influence of the pheromone (length dimension). Note that the dimension of $\beta'_i$ (number of ants$^{-1}$) differs from $\beta_i$ (number of ants$^{-1} \times$ time$^{-1}$). Thus, we write the rate of leaving the trail leading to food source $i$ as

$$S(d_i) = \frac{sd_i}{K + (\gamma_3/d_i)\beta'_i X_i}, \tag{2.5}$$

with $s$ the per capita rate of ants leaving the trail per distance unit. The term $K$ accounts for possible persistence or inertia effects which could be the result of, for instance, the memory of food location over winter or the establishment of physical trails over time improving efficiency and stability of chemical trails [12,14]. When there are very few foragers, a constant proportion $\sim sd_i/K$ of them are lost from the trail. As the number of foragers on the trail increases, this proportion decreases. Finally, when there are very many foragers, a constant number of foragers is lost, $\sim sd_i$. That is, the trail becomes more effective at retaining foragers as traffic increases, but that increased effectiveness saturates for high traffic levels. The pheromone-recruitment model including distance $d_i$ between the nest and a food source $i$ can thus be formulated as

$$\frac{dX_i}{dt} = \left( \alpha e^{-\gamma_1 d_i} + \frac{\gamma_2}{d_i} \beta_i X_i \right) \left( N - \sum_{j=1}^{J} X_j \right)$$
$$- \frac{sd_i}{K + (\gamma_3/d_i)\beta'_i X_i} X_i, \tag{2.6}$$

where $\gamma_1$, $\gamma_2$ and $\gamma_3$, are weighting coefficients, respectively, of inverse length, length and length dimensions. These three parameters control the effect of distance on scouting, and pheromone following behaviour in recruitment and trail departures, allowing the model to potentially describe the behaviour of a variety of ant species (electronic supplementary material, figure S3). Dimensions of all variables are reported in the electronic supplementary material, table S1. To validate our model in the absence of empirical data regarding the individual-scale dynamics of a system in which distance and quality are varied, we developed an agent-based model of foraging ants compatible with existing knowledge of pheromone recruitment (see the electronic supplementary material, section A). The set of assumptions of our pheromone-recruitment model is supported by the agent-based model, which predicts similar qualitative dynamics (electronic supplementary material, section C and figure S4). Namely, both models predict the same effects of colony size (trail formation is faster in larger colonies), quality (trail formation is faster and final traffic higher with food sources of higher quality) and distance (trails are more difficult to form when distance increases). This demonstrates that the differential-equation model proposed here successfully captures the qualitative dynamics generated by well-studied spatially explicit processes of individual exploration, recruitment and leaving the trail when resources differ in both quality and distance.

We ran simulations of the pheromone-recruitment model in a simple configuration with one central nest and five food sources (electronic supplementary material, section B). The quality and the distance between food sources and the nest are both uniformly distributed ($\mathcal{U}(0,20)$ and $\mathcal{U}(0,55)$, respectively). To evaluate the decentralized choice of ants at the colony level from the dynamics of trail formation in our pheromone-recruitment model, the state after 5000 time steps is saved and the average distance and average quality weighted by the number of ants committed to the trails are monitored for each simulation. We find that: (i) the distribution of the weighted average distances to exploited food sources is exponential (figure 1a); and (ii) the distribution of the weighted average quality of exploited food sources is biased towards better quality food sources compared to the initial distribution (figure 1b). Qualitatively similar results were observed for another set of parameters (electronic supplementary material, figure S3). These results show that in this model, ants successfully address the trade-off between distance and quality by minimizing the distance to food sources, while simultaneously selecting for higher quality on average. As a result, short trails are favoured but long trails can still persist if the food source is valuable to the nest (figure 1c). To optimize the allocation of workers by considering both quality of resources and distance thus results in a geometric distribution of the rank of the distance of the chosen sources (electronic supplementary material, figure S2).

To assess the hypothesis that polydomous networks are generated by pheromone-recruitment foraging mechanisms, we next compared the properties of foraging trails generated by our model to those found in empirical red wood ant networks. Polydomous ant colony networks formed by a range of species are not well described by rules minimizing the total length of trails, for instance by being connected only to nearest nodes (i.e. nests or food patches); instead, there are additional long trails connecting distant nests together [8].

*Proc. R. Soc. B* **288**: 20210430

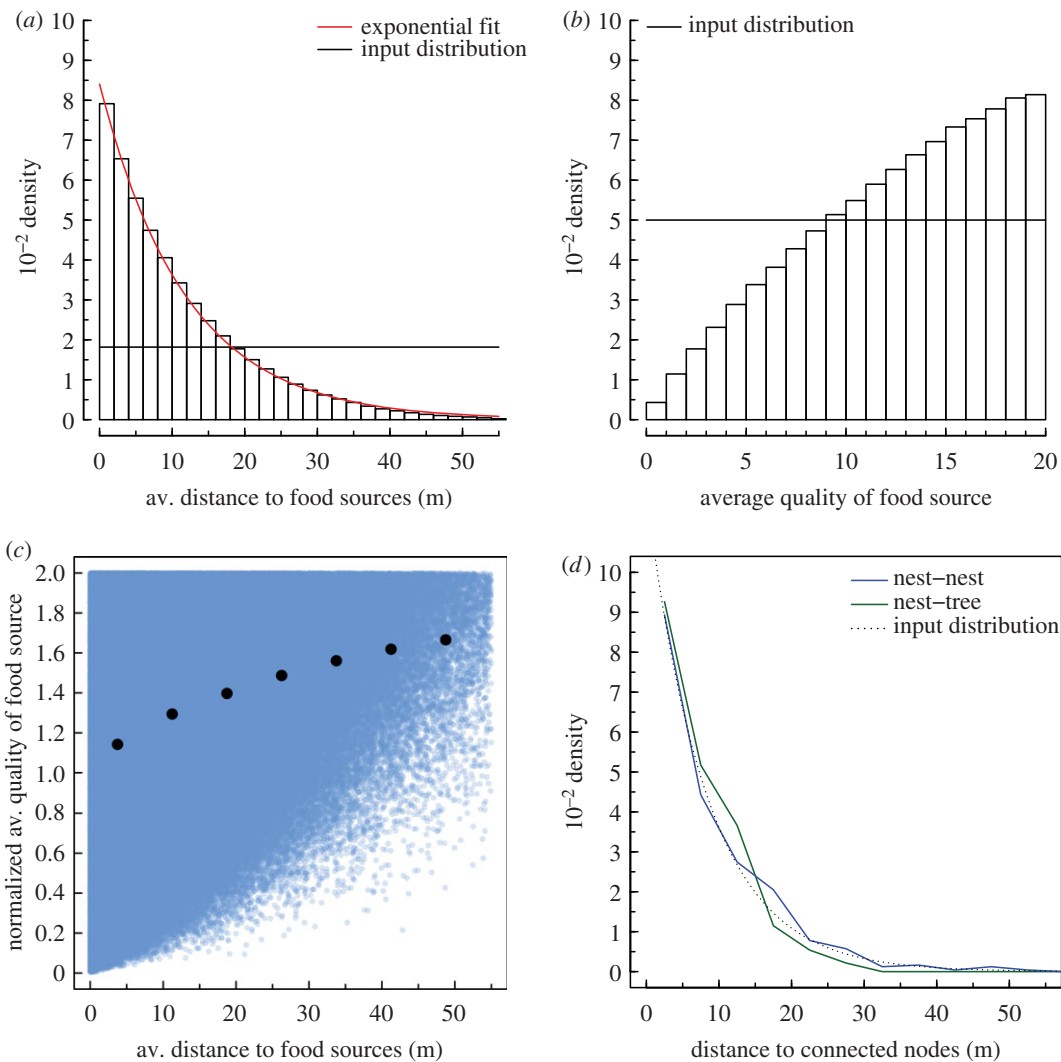

**Figure 1.** (*a–c*) Results of the 500000 simulation runs of the pheromone-recruitment model (equation 2.6) with parameters $\alpha = 0.75$ d$^{-1}$, $\gamma_1 = 0.2$ m$^{-1}$, $\gamma_2 = 0.021$ m, $\gamma_3 = 0.021$ m, $s = 3.5$ m$^{-1}$ d$^{-1}$, $K = 1$, $\eta = 20$ mol$^{-1}$ ant$^{-1}$ d$^{-1}$, $\eta' = 20$ mol$^{-1}$ ant$^{-1}$ and $N = 10000$. Distributions of the average distance (*a*) and quality (*b*) of exploited food sources weighted by the number of recruited ants (bars). Black lines are the available food source input distributions, the red line is a fitted exponential distribution, bars are simulation results. (*c*). Weighted average of the distance as a function of the weighted average of the quality for exploited food sources (black dots, mean ± s.e. of quality for binned distances). Blue dots show the result of each simulation run. (*d*) Empirical distribution of lengths of trails in *F. lugubris* polydomous networks. Dotted line shows the exponential distribution used as input in the morphogenesis model. (Online version in colour.)

By analysing an extensive field dataset collected over 7 years on red wood ants, *Formica lugubris* (see the electronic supplementary material, section F for empirical method details), we show that wood ants do not always make connections with the closest nest only (electronic supplementary material, figure S8a) and that the length of the trails is exponentially distributed (figure 1*d*). This last result shows that, in general, wood ants favour short trails but that long trails can persist. This is true for foraging trails connecting a nest with a food source, and also for internest trails connecting a pair of nests. The distributions of both lengths of internest and of foraging trails are very similar (figure 1*d*), suggesting a common underlying mechanism, in line with a former finding that polydomous ants treat other nests of their colony in the same way as food sources [15]. This underlying mechanism results in exponentially distributed trail lengths, as found in our pheromone-recruitment model (figure 1*b*); this congruence lends support to the idea that recruitment with positive feedback is the mechanism in question. While the pattern of favouring short trails plus additional long trails connecting distant nests has been observed from snapshots of networks across several species [8], we show here that it

is a property that persists over time and accounts for its origin. We show that recruitment with positive feedback used in foraging is compatible with the distributions of trail lengths found in polydomous networks. Yet, are these simple individual behavioural mechanisms sufficient to predict the structure of polydomous networks in red wood ants?

## 3. Network morphogenesis model (methods and results)

To address this question, we developed a generative colony-level model of network morphogenesis which uses a set of simple mechanistic rules compatible with the hypotheses and predictions of our pheromone-recruitment model of foraging dynamics.

This model builds networks following four assumptions:

(i) nodes (nests and trees) are spatially set in sequence, with the distance between sequential nodes drawn from the empirical exponential distributions of the distance between connected nodes (figure 1*d*);

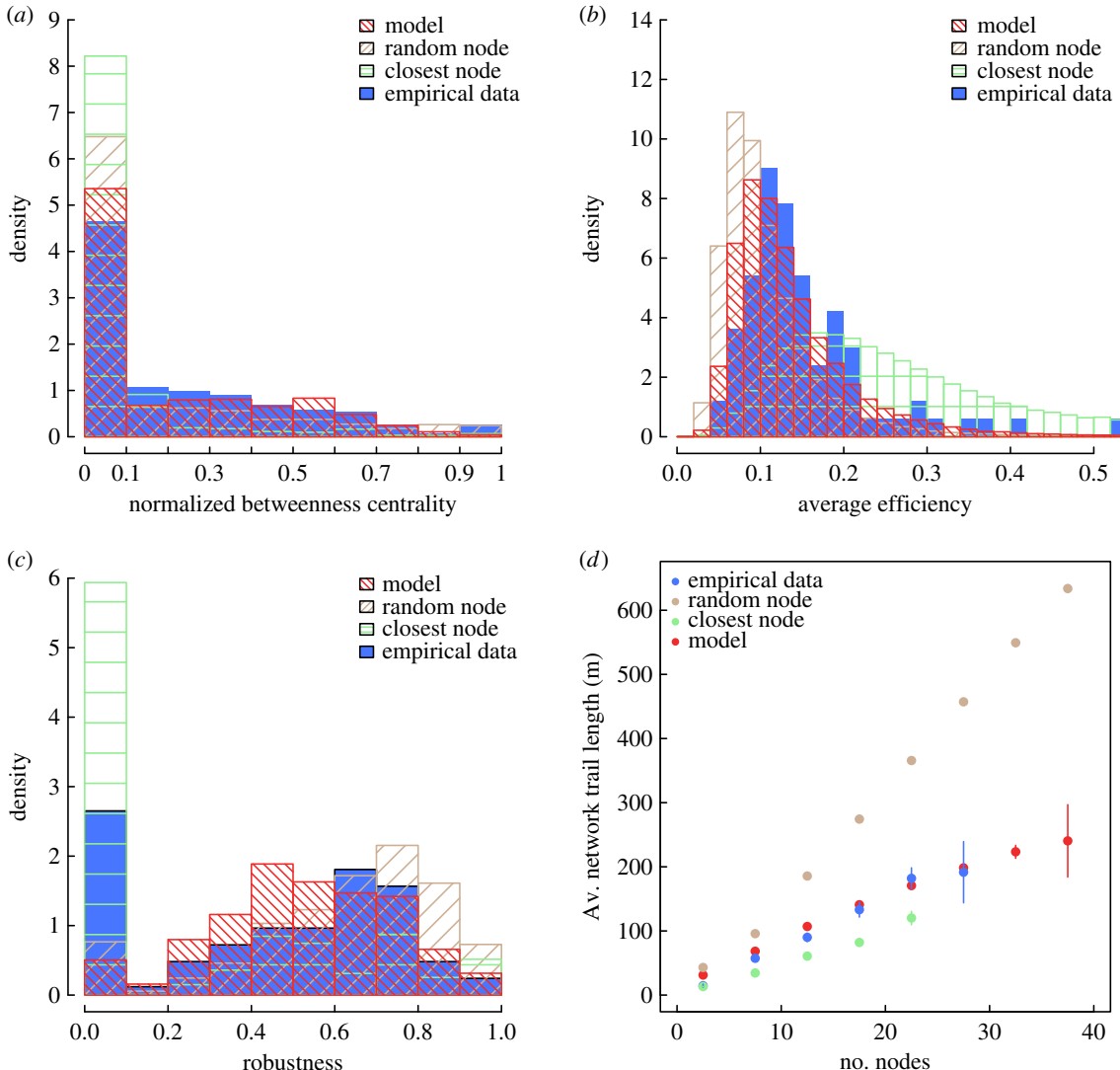

**Figure 2.** Network properties in empirical and simulated networks. Distributions (*a*) of the betweenness centrality normalized by the size of the network, (*b*) of the average efficiency, and (*c*) of network robustness in empirical data (blue histograms) and in the model (red histograms). (*d*) Average total length of trails in empirical (blue dots) and in simulated (red dots) networks. Bars show standard error. In (*a–d*), 'random node' (brown bars and dots) and 'closest node' (green bars and dots) refer to control simulations where the rule to connect nodes is respectively set to random connections and connections with the closest nodes. Statistics computed on 100000 simulated networks for each model setting. (Online version in colour.)

(ii) once all nodes have been set, connections between nodes rely on a distance-based rule and are set from the empirical geometrical distributions (electronic supplementary material, figure S8a and c), in agreement with the predictions of the pheromone-recruitment model;

(iii) each nest draws a few connections with other nodes, in line with the predictions of our pheromone-recruitment model. Namely, each nest draws only one internest connection (but can accept additional connections from other nests) and a few connections to trees following the empirical distribution (electronic supplementary material, figure S8b); and

(iv) there is no trail intersection, in agreement with empirical observations.

We simulate the generation of 100 000 networks (electronic supplementary material, figure S7) and investigate their structural properties (electronic supplementary material, section E). We compare structural properties of simulated and empirical networks collected over 7 years on red wood ants, *F. lugubris* (see the electronic supplementary material, section F for empirical method details), by evaluating a combination of network features encompassing nest centrality, network average efficiency, robustness and cost, following established approaches [8–10,17]. We measure nest centrality using betweenness centrality, which is the total number of shortest paths between pairs of nests in the network which pass through a particular nest. Network average efficiency is the average of the inverse shortest path lengths between any pair of nodes (tree or nest). Network robustness is defined as the proportion of edges that can be removed from a network without disconnecting the network (following [8]). We evaluate network cost by looking at the relationship between the total trail length and the number of nodes (nests and trees) in the networks. We also consider other emergent metrics such as the distance between nests and the number of internest trails per nest.

We find that the betweenness centrality of the simulated networks resembles the empirical distribution (figure 2*a*). Most nodes in polydomous networks have a low betweenness centrality. The networks generated by the model show a similar distribution of average efficiency, compared to the

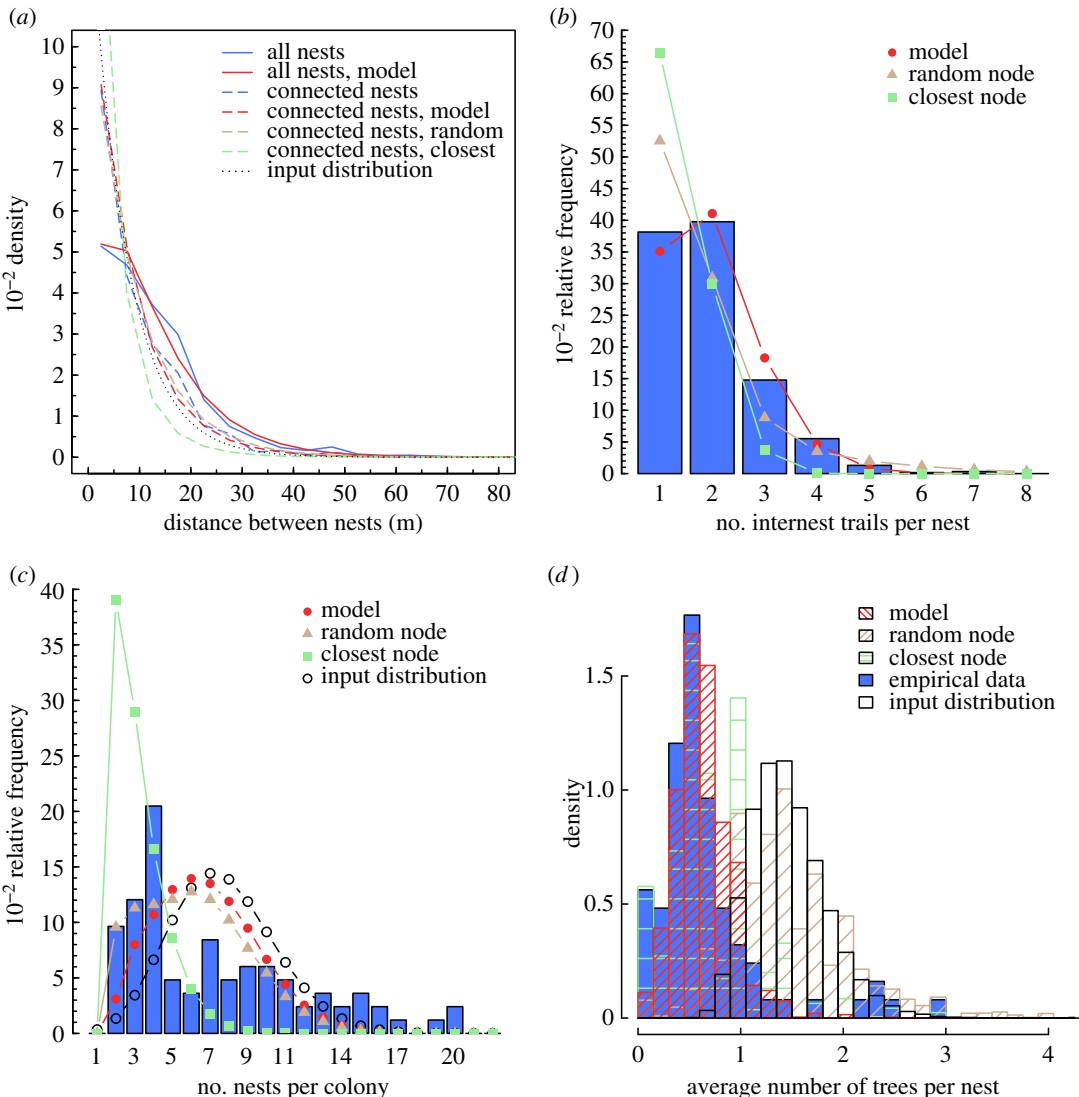

**Figure 3.** Structural properties of empirical and simulated networks. (*a*) Empirical distributions of distances to nests (solid blue) and connected nests only (dashed blue). Solid red and dashed red lines correspond to the same distributions from the simulations of our morphogenesis model. The grey dotted line shows the distribution used as input in the model. Empirical distributions (bars) of (*b*) the number of internest trails per nest and (*c*) nests per network. Red dots show the predictions of our morphogenesis model. (*d*) Distributions of the average number of trees per nest, in empirical data (blue histogram) and in simulated networks (red histogram). In (*a*–*d*), 'random node' (brown lines, dots and bars) and 'closest node' (green lines, dots and bars) refer to alternate models which only differ in their rules to connect nodes, respectively set to random connections and connections with the closest nodes. Statistics computed on 100000 simulated networks for each model setting. (Online version in colour.)

empirical networks: in both cases, most networks are relatively inefficient, but there is a long tail of more efficient networks (figure 2*b*). The empirically determined average path length and network connectivity control the mode in average efficiency, while the empirically determined minimum inter-node distance places an upper limit on average efficiency. For network robustness $R$, we find that both empirical and simulated networks have on average a robustness close to 0.5 and distributed between 0.2 and 0.8 (figure 2*c*). There is however a discrepancy between empirical and simulated networks, with the empirical network having a peak at $R = 0$ not found in simulated networks. A null robustness indicates a minimum spanning tree network structure, in which any edge removal results in disrupting the network global connection. The presence of a peak at $R = 0$ in empirical data might be a result of warm and dry weather in recent years (see the electronic supplementary material, section E2). We find overall that the network cost, as represented by the relationship between the total length of trails and the number of nodes, is very similar in empirical

and simulated networks (figure 2*d*). Our morphogenesis model also captures the distribution of distances between all nests, even including those which are not connected—this is an emergent property since the input distribution used to spatially set nodes is calculated only from connected nodes (figure 3*a*). Drawing only one internest connection per nest appears to be sufficient to generate the entire degree distribution of internest trails per nest (figure 3*b*); this supports the idea that foraging mechanisms, usually favouring a small number of trails from each nest, are compatible with the morphogenesis of *F. lugubris* polydomous networks.

To assess the importance of the connection rule in our model, we ran further simulations with two other rules of node connection: a first one in which connections between available nodes are made randomly and a second one in which connections between available nodes are always made between the closest nodes (respectively, 'random node' in brown and 'closest node' in green on figures 2 and 3)—all other aspects of the model being the same. These simulations allow us to compare our model to alternative

rules where ants do not take the distance-quality trade-off into account but may still minimize transportation costs. These additional simulations show that our model outperforms both alternative rules; our model is the only one to perform well against empirical data in all the metrics analysed. When connections are made randomly, results are only slightly worse than our model regarding the nest centrality, the average network efficiency, the network robustness and the number of nests per colony (figures 2a–c and 3c), but are substantially worse regarding the cost and the average number of trees per nest (figures 2d and 3d). The model with connections made only to closest nodes performs worse than our model across all metrics. The poor performance of these alternative rules underscores the importance of a rule of node connection based on the distance-quality trade-off in generating networks similar to empirical ant polydomous networks.

# 4. Discussion

We have shown that simulated networks of a morphogenesis model are consistent with ecologically significant structural and geometric properties of empirical red wood ant polydomous networks. Moreover, using a pheromone-recruitment model, we show that the underlying assumptions of our morphogenesis model are compatible with mechanisms of recruitment with positive feedback. These findings suggest that common coordination and decision-making mechanisms might govern the morphogenesis, the growth and the dynamics of polydomous transport networks in ant colonies. The co-option of a pre-existing essential behaviour (foraging) might explain why polydomy has evolved several times in ants [7]. Positive feedback mechanisms, balancing quality and spatial factors, could be one of the key elements of a self-organizing process leading to the morphogenesis of polydomous networks. In agreement with empirical data and theoretical predictions, we show that positive feedback mechanisms favour the exploitation of a few, close and high-quality sources. In polydomous networks, however, trails connect not only nests to food sources but also nests to other nests. Ants in polydomous colonies can treat other nests similarly to food sources [15]; our results suggest that similar processes underlie the emergence of both foraging and internest trails. There is clearly a trade-off between (potential) quality of resources and distance to these resources that mechanisms of recruitment successfully address—ultimately, this may be an essential element shaping the structure of polydomous networks. Our study suggests that mechanisms of recruitment could be under additional selection pressures in polydomous ants, since they not only control foraging activity but potentially also the cost, efficiency and robustness of transporting resources between nests. To better understand how differing ecological pressures may shape the evolution of polydomy in ants, future work could evaluate the quantitative predictions of pheromone-recruitment models for network performance metrics across species that differ in foraging dynamics and network structures.

Our work represents a first step in developing a theory for the structure of biological multi-source multi-sink transport networks. It highlights both the potential of generative models relying on explicit behavioural mechanisms in reaching this goal as well as the suitability of the polydomous ant network system to investigate mechanisms of the formation of transport networks. Our research suggests that polydomous ant networks can be generated via a sequence of behaviour consisting of (i) an exploratory phase to discover potential resource sources, and (ii) a selection phase to establish trails towards the best sources based on a positive feedback mechanism. This sequence is common to self-organized biological networks found for instance in fungi [2] or slime moulds [5,35]. We note that the networks of the polydomous ant system mapped in the field rarely show any trail intersections, so are planar graphs without Steiner points [8], unlike networks of polydomous ants collected under laboratory conditions [36] or networks of slime moulds [5]. The study of the consequences of the absence of trail intersections and Steiner points in the network structure and properties, for instance by using the modelling framework developed in this article, may help to establish further general principles of self-organized biological networks. While the topology of the resulting networks differs between the ant, fungal and slime mould systems, the striking resemblance of their underlying mechanisms indeed suggest a unifying theory of self-organized biological networks based on the combination of exploration and positive reinforcement of best sources. Such a unifying theory could have broad practical applications for generating networks with different properties under various conditions.

Data accessibility. Code implementing the agent-based model and the model of network morphogenesis is available on figshare https://doi.org/10.6084/m9.figshare.c.5309864.

Authors' contributions. V.L., M.C.D.M. and E.J.H.R. designed research; V.L. performed research; H.L. contributed to developing the individual-based model; S.E., D.D.R.B. and E.J.H.R. provided the empirical data; V.L. analysed data with feedback from S.P., M.C.D.M. and E.J.H.R.; V.L. wrote the paper with the assistance of all authors.

Competing interests. The authors declare no competing interests.

Funding. This work was funded by NSF award IOS 1755425 (V.L., M.C.D.M. and E.J.H.R) and NSF award IOS 1755406 (S.P.). The field data were collected under studentships funded by the NERC ACCE DTP (grant code: NE/L002450/1) and the National Trust.

Acknowledgements. We thank the National Trust for allowing the fieldwork to be performed on their land. We also thank three anonymous reviewers for their supportive and considered comments during the review process.

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
