## [Peer Review File · Proceedings of the Royal Society B: Biological Sciences]

Review History

RSPB-2020-2760.R0 (Original submission)

Review form: Reviewer 1

Recommendation

Major revision is needed (please make suggestions in comments)

Scientific importance: Is the manuscript an original and important contribution to its field?

Excellent

General interest: Is the paper of sufficient general interest?

Good

Quality of the paper: Is the overall quality of the paper suitable?

Excellent

Is the length of the paper justified?

Yes

Should the paper be seen by a specialist statistical reviewer?

No

Do you have any concerns about statistical analyses in this paper? If so, please specify them explicitly in your report.

Yes

It is a condition of publication that authors make their supporting data, code and materials available - either as supplementary material or hosted in an external repository. Please rate, if applicable, the supporting data on the following criteria.

Is it accessible?

Yes

Is it clear?

Yes

Is it adequate?

No

Do you have any ethical concerns with this paper?

No

Comments to the Author

Overview:

The authors describe their approach for modelling the generation of multi-source multi-sink networks inspired by the collective behaviour of ants. They find that the well-studied phenomenon of recruitment via positive feedback is pretty much all that is necessary to generate networks that are similar to ant networks. This finding in itself does not sound particularly novel or exciting - but there is much more to this paper than that, and even the authors are risking underselling themselves at some points in the manuscript.

It may seem obvious that positive feedback via pheromone-based recruitment is important for maintaining the internet networks of ants that largely rely on pheromone-based recruitment for foraging. The value of this paper is in taking the rare approach of building up from individual-level first principles to demonstrate how networks are built and maintained at the group-level. This is not just an exercise in model fitting and validation - the authors build their underlying pheromone-recruitment model from a broad range of empirical data, combine this with a network morphogenesis model that itself draws from the parameter space defined by empirical data from wood ants, to produce networks that are validated against the real world networks. It is an elegant demonstration of how emergent group-level properties arise from simple individual-level behaviour. More than that, it is generalisable to more than just the study of wood ant nests. These two key points make the article well worth publishing in Proceedings B, but really should be emphasised in revisions of the manuscript.

The walkthrough of the model design and description of equations was extremely well done - some of the clearest explanations we have seen - and the authors should be commended for their writing, especially here, but also throughout the manuscript.

There are some issues with how the manuscript is arranged, and many missing details for certain aspects of the models, the presentation of results and the significance/discussion of various key features being overlooked. As such, we recommend publication after the listed issues are addressed under a major revision.

Major points:

A major flaw in this manuscript is the arrangement of its presentation - virtually everything of importance is hidden in the ESM, while much less important features are presented in the main text. We have pointed several issues out below, but in general, Figures 1 and 3 should be ESM, while at least Figs S2, S6 and many points of description in the ESM should be presented in the main text.

Title: You should consider changing the title to better highlight the novelty of your results. As is, it risks underselling your study.

Main text:

Figure 1 appears to serve as an overview of the authors design approach, but this information is already supplied in the text. I would consider removing this figure entirely, or at least exchanging it with one of the more useful figures in the ESM. Also, at it's current position the content in Fig 1 appears long before the reader knows what each of the mentioned studies and models are. The same comment applies to Fig 3 - this Figure doesn't add much value to the manuscript, and could easily be swapped for more informative info in the ESM. So the suggested removal/exchange would add to both clarity and relevance.

Lines 153-156: You need to do more to justify the addition of K - as it stands it seems like it could be a useful fudge factor that aids the fitting of your model. It may be entirely appropriate to include "possible persistence or inertia effects" in the model, but why? What references (and ideally, data) support this addition? Prove to me that this isn't just a free parameter to help you fit your model. Or if it is a useful free parameter, explain why it is necessary, and the impact its addition has on the model.

Line 169: I'm not convinced that the two models are so similar - you point out some general similarities, but there are some glaring disagreements that aren't discussed. For instance, the biggest disparities appear to be for mediocre food qualities placed at medium distances from the nest (the biggest difference being for large colony sizes at $q=5$, $d=0.5$ and 0.55 , but also see $q=1$, $d=0.4$). The dynamics can appear totally different between the two models, with the ABM appearing to generate stronger trails than the pheromone-recruitment model. The ABM results are averaged over 10 runs, but there is no indication of the variance of these runs. This is especially important for stochastic models with positive feedback, which can generate quite a lot of variance that can run away in the initial stages. For this reason, I am not convinced 10 runs is enough to make an accurate judgement on the average behaviour of the model.

Fig 4 (and others): Much of the data is described as averages (e.g. Fig 4 a, b and c) but there is no indication of the variance. Please add error bars or 95% confidence intervals where necessary. This has been done for 4d. In 4d, why is there no corresponding empirical data for the 30 and 35 node positions on the x-axis? Perhaps this is an extension of the model beyond the size of the natural networks you could observe? If so, please discuss, especially why it predicts a decrease in average network trail length (perhaps this is an error here, as the red point for 30 nodes has error bars, while none of the other data points for the model do?).

There is no statistical analysis of the closeness of the model fit to the empirical observations. While a qualitative comparison is fine, a statistical test of some kind would be instructive (especially in the case of 4c).

ESM:

Line 100: Is the decision not to allow trail intersections based on empirical evidence? If so, please discuss. I feel this is important because network intersections, especially steiner points, are important features of many biological networks;

Latty, T., Ramsch, K., Ito, K., Nakagaki, T., Sumpter, D.J., Middendorf, M. and Beekman, M., 2011. Structure and formation of ant transportation networks. *Journal of The Royal Society Interface*, 8(62), pp.1298-1306.

Liu, L., Song, Y., Zhang, H., Ma, H. and Vasilakos, A.V., 2013. Physarum optimization: A biology-inspired algorithm for the steiner tree problem in networks. *IEEE Transactions on Computers*, 64(3), pp.818-831.

Tero, A., Takagi, S., Saigusa, T., Ito, K., Bebbler, D.P., Fricker, M.D., Yumiki, K., Kobayashi, R. and Nakagaki, T., 2010. Rules for biologically inspired adaptive network design. *Science*, 327(5964), pp.439-442.

If you are "developing a theory for the structure of biological multi-source multi-sink transport

networks" (lines 318-319), this could become an important issue.

Minor points:

Main text:

Line 49: Can you clarify what you mean by "the scale of the nests?" Do you mean the colony level, or some intermediate level between individual and colony?

Lines 65-67: "Multi-source multi-sink" appears a couple of times within to sentences. Consider rephrasing?

Line 99: q_i appears in Eq 1, but is not described until Eq 2. Please describe after Eq 1. Also, q_i is described as the quality of the source i , but in Table S1, it is listed as the "Amount of substance". If amount of substance is directly proportional to the quality of the source i , this needs to be described more clearly.

Line 164: compensate for the

Line 164: We were able to find one paper (Fewell et al., 1992. Distance effects on resource profitability and recruitment in the giant tropical ant, *Paraponera clavata*. *Oecologia*) in which authors described recruitment behaviour in giant tropical ants when modifying both distance and quality of food sources.

Fig 2: Why no red line for exponential fit for Fig 2b?

Line 278: The explanation for this discrepancy should not be buried in the ESM, but brought here.

Line 289: Figure 7c in the ESM is not mentioned in the text. Is this because the model does not seem to match the apparent bimodal distribution of the empirical data? This should be discussed.

ESM:

Line 24: We understand that there is no need for reporting parameter space exploration for each of these. However, it would be nice to have some more information on how these values have been obtained. Are they derived from empirical data (if so, you should provide a reference) or from personal observations?

Line 44: Parameters of the pheromone-recruitment

Line 115: have not been accounted for

Figure S2: Please provide a description of the red line in the caption.

Figure S3: The caption should mention that the data you are plotting are averaged over 10 runs (as you explain in Line 42 in the ESM)

Figure S5: This figure is particularly difficult to follow. You may find a better way to plot this data (i.e., boxplots)?

Figure S6: This figure may be included in the main text. This way, you can show the distribution of the parameters used in the morphogenesis model without having to refer to the ESM.

Figure S7c: The distributions are very different from each other. You should include a note in the caption explaining the reason for such a difference, and/or provide a link to L81-82 in the ESM.

Review form: Reviewer 2

Recommendation

Major revision is needed (please make suggestions in comments)

Scientific importance: Is the manuscript an original and important contribution to its field?

Marginal

General interest: Is the paper of sufficient general interest?

Acceptable

Quality of the paper: Is the overall quality of the paper suitable?

Acceptable

Is the length of the paper justified?

Yes

Should the paper be seen by a specialist statistical reviewer?

No

Do you have any concerns about statistical analyses in this paper? If so, please specify them explicitly in your report.

Yes

It is a condition of publication that authors make their supporting data, code and materials available - either as supplementary material or hosted in an external repository. Please rate, if applicable, the supporting data on the following criteria.

Is it accessible?

Yes

Is it clear?

Yes

Is it adequate?

Yes

Do you have any ethical concerns with this paper?

No

Comments to the Author

The manuscript describes a model that implements recruitment with positive feedback as a potential mechanism for transport network morphogenesis in ants. I think that the authors present a potentially useful approach here, but the manuscript requires more work to highlight the novelty of the model introduced here and its potential as a predictive model.

There are several issues the authors need to address :

- 1) My first concern is about the supplementary material. I found a bit frustrating that essential information and results are only presented in the supplementary material. (on a slighter note, the grammar has not been reviewed carefully throughout the supplementary material - see below for detailed comments)
- 2) My second concerns is related to the novelty of the work presented here. The authors should better explain how the model proposed in this paper differs from previous models. It was not clear to me what was new in this paper.
- 3) My third concern regards the ABM presentation. The ABM should be introduced in the manuscript and not only in the SI. In addition, the authors should be more explicit regarding the parameters and assumptions of their ABM (see below for more detailed comments)

Detailed comments :

Line 90-91 : "the idea that the presence of an ant on a trail can be seen at the colony level as a vote of the individual in favour of the resource connected to the trail."

=> not entirely sure that this analogy is needed here

Line 106-108 : "To examine how the quality of a food source and its distance from the nest interact to influence ant foraging, we reformulate the Sumpter and Beekman model to make all three mechanisms dependent on distance."

=> should be "To examine how the quality of a food source and its distance from the nest interact to influence ant foraging, we reformulate the Sumpter and Beekman model to make all three mechanisms dependent on distance AND QUALITY"

Line 115 'by analysing available published data of single ants *Temnothorax albipennis* exploring the area around their nest entrance'

=> Why using published data of single ants *Temnothorax albipennis* exploring the area around their nest entrance, I don't believe this species builds multi-sink multi-source transport networks. Could the authors better justify their choice?

Line 117-119 : Lack of clarity, the sentence should be rewritten.

Line 127-129: We also assume that recruitment mechanisms mainly rely on pheromone trails, at least in the emergence phase of the trail – other processes possibly being involved after the trail formation, such as workers remembering the location of the food source.

=> something is missing in this sentence.

line 138-151: this whole section needs to be re-written to be more explanatory

Line 159: the authors should better introduced the weighting parameters. What is the sensitivity of these parameters ? How was the model calibrated ?

The authors never refers to figure S7c. (or I missed it !)

Figure 3: I don't see how this figure can help the reader. It is by far more interesting to have Fig S6 and S7 in the manuscript instead of this figure. Figure 3 could be moved in the SI.

Line 269-291 : all comparisons are qualitative, is there a way to quantify if the model properly fit the data.

Line 326: "While the topology and the functions of the resulting networks differ between the ant, fungal and slime mould systems"

=> The function of fungal and slime mould networks is also to transport resources, no? The topology is indeed different, but the function remains the same.

Supplementary material :

Line 16-17: "When a food source is found, the successful ant returns to the nest and deposits a quantity of pheromones which is proportional to the quality of the food source."

=> what path the ant is following when it returns to the nest?

Line 19-20 : When these cells have pheromones, ants follow the gradient of pheromone concentration by stochastically choosing the cell with the highest concentration with probability $p = 0.95$.

=> this probability seems quite high, could you justify such a high value empirically

Line 20-21 : "At each time step without pheromone in the three cells ants face, they have a set probability, constant over time, to return to the nest."

=> what is the value of this probability

Fig S1 : "Distribution of the rank of distance of the source selected by the colony, i.e. the source with the most crowded trail (bars), and fitted theoretical geometric distribution (dots) in simulations of the pheromone-recruitment model (parameters in Methods)."

=> please rephrase for clarity

Fig S2 : "for ants less efficient in pheromone communication "

=> could you be more explicit here.

Fig S2: the legend is illegible for the first panels, the authors should move it to the bottom right corner

Here are some grammar errors I spotted in the SI :

Fig S6 : "Number of nest per colony" (x)-axis
=> should be "Number of nestS per colony"

Fig S6 : "(d) respectively show the distributions of the number of nest per colony"
=> should be "number of nestS per colony"

Fig 7: "Grey dotted line show the distribution used as input in the model."
=> should be "Grey dotted line showS the distribution used as input in the model"

Fig 7b: "Number of internest trail per nest" (x)-axis
=> should be "Number of internest trailS per nest"

Fig 7c: "Number of tree" (x)-axis
=> should be "Number of treeS"

Decision letter (RSPB-2020-2760.R0)

15-Dec-2020

Dear Dr Lecheval:

I am writing to inform you that your manuscript RSPB-2020-2760 entitled "Complex biological transport networks are explained by recruitment with positive feedback" has, in its current form, been rejected for publication in Proceedings B.

This action has been taken on the advice of referees, who have recommended that substantial revisions are necessary. With this in mind we would be happy to consider a resubmission, provided the comments of the referees are fully addressed. However please note that this is not a provisional acceptance.

Sincerely,
 Professor Gary Carvalho
 mailto: proceedingsb@royalsociety.org

Associate Editor

Board Member: 1

Comments to Author:

Your manuscript has now been seen by two expert reviewers (please note, one review was done by both the invited reviewer and a PhD student in their group, hence the use of the 'we' pronoun in their review). There is a stark contrast between the reviewers with respect to the novelty of your manuscript - one believes you are underplaying it, whilst the other cannot see the novelty. I would like to give you the opportunity to convince this latter reviewer (which in turn should address the underplaying issue raised by the former). Consequently, in a revised manuscript and cover letter, you must provide a convincing argument as to the novelty of your modeling with respect to publication in this journal. In addition, both reviewers have recommended significant reshuffling of content between the main ms. and the supplementary materials, and the necessity for statistical analyses of model fitting, and I view both of these recommendations as requirements. There are also numerous other comments that you would need to address in a revision. Please remember that if the reviewers find something problematic, it is not enough to address it in a cover letter - the manuscript needs to change commensurately too.

Reviewer(s)' Comments to Author:

Referee: 1

Comments to the Author(s)

Overview:

The authors describe their approach for modelling the generation of multi-source multi-sink networks inspired by the collective behaviour of ants. They find that the well-studied phenomenon of recruitment via positive feedback is pretty much all that is necessary to generate networks that are similar to ant networks. This finding in itself does not sound particularly novel or exciting - but there is much more to this paper than that, and even the authors are risking underselling themselves at some points in the manuscript.

It may seem obvious that positive feedback via pheromone-based recruitment is important for maintaining the internet networks of ants that largely rely on pheromone-based recruitment for foraging. The value of this paper is in taking the rare approach of building up from individual-level first principles to demonstrate how networks are built and maintained at the group-level. This is not just an exercise in model fitting and validation - the authors build their underlying pheromone-recruitment model from a broad range of empirical data, combine this with a network morphogenesis model that itself draws from the parameter space defined by empirical data from wood ants, to produce networks that are validated against the real world networks. It is an elegant demonstration of how emergent group-level properties arise from simple individual-level behaviour. More than that, it is generalisable to more than just the study of wood ant nests. These two key points make the article well worth publishing in Proceedings B, but really should be emphasised in revisions of the manuscript.

The walkthrough of the model design and description of equations was extremely well done - some of the clearest explanations we have seen - and the authors should be commended for their writing, especially here, but also throughout the manuscript.

There are some issues with how the manuscript is arranged, and many missing details for certain aspects of the models, the presentation of results and the significance/discussion of various key

features being overlooked. As such, we recommend publication after the listed issues are addressed under a major revision.

Major points:

A major flaw in this manuscript is the arrangement of its presentation – virtually everything of importance is hidden in the ESM, while much less important features are presented in the main text. We have pointed several issues out below, but in general, Figures 1 and 3 should be ESM, while at least Figs S2, S6 and many points of description in the ESM should be presented in the main text.

Title: You should consider changing the title to better highlight the novelty of your results. As is, it risks underselling your study.

Main text:

Figure 1 appears to serve as an overview of the authors design approach, but this information is already supplied in the text. I would consider removing this figure entirely, or at least exchanging it with one of the more useful figures in the ESM. Also, at it's current position the content in Fig 1 appears long before the reader knows what each of the mentioned studies and models are. The same comment applies to Fig 3 – this Figure doesn't add much value to the manuscript, and could easily be swapped for more informative info in the ESM. So the suggested removal/exchange would add to both clarity and relevance.

Lines 153-156: You need to do more to justify the addition of K – as it stands it seems like it could be a useful fudge factor that aids the fitting of your model. It may be entirely appropriate to include “possible persistence or inertia effects” in the model, but why? What references (and ideally, data) support this addition? Prove to me that this isn't just a free parameter to help you fit your model. Or if it is a useful free parameter, explain why it is necessary, and the impact its addition has on the model.

Line 169: I'm not convinced that the two models are so similar – you point out some general similarities, but there are some glaring disagreements that aren't discussed. For instance, the biggest disparities appear to be for mediocre food qualities placed at medium distances from the nest (the biggest difference being for large colony sizes at $q=5$, $d=0.5$ and 0.55 , but also see $q=1$, $d=0.4$). The dynamics can appear totally different between the two models, with the ABM appearing to generate stronger trails than the pheromone-recruitment model. The ABM results are averaged over 10 runs, but there is no indication of the variance of these runs. This is especially important for stochastic models with positive feedback, which can generate quite a lot of variance that can run away in the initial stages. For this reason, I am not convinced 10 runs is enough to make an accurate judgement on the average behaviour of the model.

Fig 4 (and others): Much of the data is described as averages (e.g. Fig 4 a, b and c) but there is no indication of the variance. Please add error bars or 95% confidence intervals where necessary. This has been done for 4d. In 4d, why is there no corresponding empirical data for the 30 and 35 node positions on the x-axis? Perhaps this is an extension of the model beyond the size of the natural networks you could observe? If so, please discuss, especially why it predicts a decrease in average network trail length (perhaps this is an error here, as the red point for 30 nodes has error bars, while none of the other data points for the model do?).

There is no statistical analysis of the closeness of the model fit to the empirical observations. While a qualitative comparison is fine, a statistical test of some kind would be instructive (especially in the case of 4c).

ESM:

Line 100: Is the decision not to allow trail intersections based on empirical evidence? If so, please discuss. I feel this is important because network intersections, especially steiner points, are important features of many biological networks;

Latty, T., Ramsch, K., Ito, K., Nakagaki, T., Sumpter, D.J., Middendorf, M. and Beekman, M., 2011. Structure and formation of ant transportation networks. *Journal of The Royal Society Interface*, 8(62), pp.1298-1306.

Liu, L., Song, Y., Zhang, H., Ma, H. and Vasilakos, A.V., 2013. Physarum optimization: A biology-inspired algorithm for the steiner tree problem in networks. *IEEE Transactions on Computers*, 64(3), pp.818-831.

Tero, A., Takagi, S., Saigusa, T., Ito, K., Bebber, D.P., Fricker, M.D., Yumiki, K., Kobayashi, R. and Nakagaki, T., 2010. Rules for biologically inspired adaptive network design. *Science*, 327(5964), pp.439-442.

If you are “developing a theory for the structure of biological multi-source multi-sink transport networks” (lines 318-319), this could become an important issue.

Minor points:

Main text:

Line 49: Can you clarify what you mean by “the scale of the nests?” Do you mean the colony level, or some intermediate level between individual and colony?

Lines 65-67: “Multi-source multi-sink” appears a couple of times within to sentences. Consider rephrasing?

Line 99: q_i appears in Eq 1, but is not described until Eq 2. Please describe after Eq 1. Also, q_i is described as the quality of the source i , but in Table S1, it is listed as the “Amount of substance”. If amount of substance is directly proportional to the quality of the source i , this needs to be described more clearly.

Line 164: compensate for the

Line 164: We were able to find one paper (Fewell et al., 1992. Distance effects on resource profitability and recruitment in the giant tropical ant, *Paraponera clavata*. *Oecologia*) in which authors described recruitment behaviour in giant tropical ants when modifying both distance and quality of food sources.

Fig 2: Why no red line for exponential fit for Fig 2b?

Line 278: The explanation for this discrepancy should not be buried in the ESM, but brought here.

Line 289: Figure 7c in the ESM is not mentioned in the text. Is this because the model does not seem to match the apparent bimodal distribution of the empirical data? This should be discussed.

ESM:

Line 24: We understand that there is no need for reporting parameter space exploration for each of these. However, it would be nice to have some more information on how these values have been obtained. Are they derived from empirical data (if so, you should provide a reference) or from personal observations?

Line 44: Parameters of the pheromone-recruitment

Line 115: have not been accounted for

Figure S2: Please provide a description of the red line in the caption.

Figure S3: The caption should mention that the data you are plotting are averaged over 10 runs (as you explain in Line 42 in the ESM)

Figure S5: This figure is particularly difficult to follow. You may find a better way to plot this data (i.e., boxplots)?

Figure S6: This figure may be included in the main text. This way, you can show the distribution of the parameters used in the morphogenesis model without having to refer to the ESM.

Figure S7c: The distributions are very different from each other. You should include a note in the caption explaining the reason for such a difference, and/or provide a link to L81-82 in the ESM.

Referee: 2

Comments to the Author(s)

The manuscript describes a model that implements recruitment with positive feedback as a potential mechanism for transport network morphogenesis in ants. I think that the authors present a potentially useful approach here, but the manuscript requires more work to highlight the novelty of the model introduced here and its potential as a predictive model.

There are several issues the authors need to address :

1) My first concern is about the supplementary material. I found a bit frustrating that essential information and results are only presented in the supplementary material. (on a slighter note, the grammar has not been reviewed carefully throughout the supplementary material - see below for detailed comments)

2) My second concerns is related to the novelty of the work presented here. The authors should better explain how the model proposed in this paper differs from previous models. It was not clear to me what was new in this paper.

3) My third concern regards the ABM presentation. The ABM should be introduced in the manuscript and not only in the SI. In addition, the authors should be more explicit regarding the parameters and assumptions of their ABM (see below for more detailed comments)

Detailed comments :

Line 90-91 : "the idea that the presence of an ant on a trail can be seen at the colony level as a vote of the individual in favour of the resource connected to the trail."

=> not entirely sure that this analogy is needed here

Line 106-108 : "To examine how the quality of a food source and its distance from the nest interact to influence ant foraging, we reformulate the Sumpter and Beekman model to make all three mechanisms dependent on distance."

=> should be "To examine how the quality of a food source and its distance from the nest interact to influence ant foraging, we reformulate the Sumpter and Beekman model to make all three mechanisms dependent on distance AND QUALITY"

Line 115 'by analysing available published data of single ants *Temnothorax albigenis* exploring the area around their nest entrance'

=> Why using published data of single ants *Temnothorax albigenis* exploring the area around their nest entrance, I don't believe this species builds multi-sink multi-source transport networks. Could the authors better justify their choice?

Line 117-119 : Lack of clarity, the sentence should be rewritten.

Line 127-129: We also assume that recruitment mechanisms mainly rely on pheromone trails, at least in the emergence phase of the trail - other processes possibly being involved after the trail formation, such as workers remembering the location of the food source.

=> something is missing in this sentence.

line 138-151: this whole section needs to be re-written to be more explanatory

Line 159: the authors should better introduced the weighting parameters. What is the sensitivity of these parameters ? How was the model calibrated ?

The authors never refers to figure S7c. (or I missed it !)

Figure 3: I don't see how this figure can help the reader. It is by far more interesting to have Fig S6 and S7 in the manuscript instead of this figure. Figure 3 could be moved in the SI.

Line 269-291 : all comparisons are qualitative, is there a way to quantify if the model properly fit the data.

Line 326: "While the topology and the functions of the resulting networks differ between the ant, fungal and slime mould systems"

=> The function of fungal and slime mould networks is also to transport resources, no? The topology is indeed different, but the function remains the same.

Supplementary material :

Line 16-17: "When a food source is found, the successful ant returns to the nest and deposits a quantity of pheromones which is proportional to the quality of the food source."

=> what path the ant is following when it returns to the nest?

Line 19-20 : When these cells have pheromones, ants follow the gradient of pheromone concentration by stochastically choosing the cell with the highest concentration with probability $p = 0.95$.

=> this probability seems quite high, could you justify such a high value empirically

Line 20-21 : "At each time step without pheromone in the three cells ants face, they have a set probability, constant over time, to return to the nest."

=> what is the value of this probability

Fig S1 : "Distribution of the rank of distance of the source selected by the colony, i.e. the source with the most crowded trail (bars), and fitted theoretical geometric distribution (dots) in simulations of the pheromone-recruitment model (parameters in Methods)."

=> please rephrase for clarity

Fig S2 : "for ants less efficient in pheromone communication "

=> could you be more explicit here.

Fig S2: the legend is illegible for the first panels, the authors should move it to the bottom right corner

Here are some grammar errors I spotted in the SI :

Fig S6 : "Number of nest per colony" (x)-axis

=> should be "Number of nestS per colony"

Fig S6 : "(d) respectively show the distributions of the number of nest per colony"

=> should be "number of nestS per colony"

Fig 7: "Grey dotted line show the distribution used as input in the model."

=> should be "Grey dotted line showS the distribution used as input in the model"

Fig 7b: "Number of internest trail per nest" (x)-axis

=> should be "Number of internest trailS per nest"

Fig 7c: "Number of tree" (x)-axis

=> should be "Number of treeS"

Author's Response to Decision Letter for (RSPB-2020-2760.R0)

See Appendix A.

RSPB-2021-0430.R0

Review form: Reviewer 1

Recommendation

Accept with minor revision (please list in comments)

Scientific importance: Is the manuscript an original and important contribution to its field?

Excellent

General interest: Is the paper of sufficient general interest?

Excellent

Quality of the paper: Is the overall quality of the paper suitable?

Excellent

Is the length of the paper justified?

Yes

Should the paper be seen by a specialist statistical reviewer?

No

Do you have any concerns about statistical analyses in this paper? If so, please specify them explicitly in your report.

No

It is a condition of publication that authors make their supporting data, code and materials available - either as supplementary material or hosted in an external repository. Please rate, if applicable, the supporting data on the following criteria.

Is it accessible?

Yes

Is it clear?

Yes

Is it adequate?

Yes

Do you have any ethical concerns with this paper?

No

Comments to the Author

The resubmitted version is a dramatic improvement, and we could suggest only a few minor improvements. We therefore recommend acceptance of this manuscript with only a few adjustments.

The authors addressed the concerns we had on the previous version, providing thorough explanations and modifying the manuscript accordingly.

The new title is more fitting of the story and better highlights the bottom-up approach of the study. All figures have been greatly improved, and so is their new arrangement between the main text and the ESM. The explanation of the K variable makes its effects and behaviour much clearer, in biological terms. The additional replication of the ABM and the presentation of the variance adds trustworthiness to the data without changing the original results.

We particularly like the models with alternative rules and their description (lines 394-410).

Minor points for correction:

Main text:

Line 204: The refence should be to "Electronic Supplementary Materials, Section C" and not to Section A.

Acknowledgements: 3 reviewers! - 'Reviewer 1' is a superorganismic co-reviewer, consisting of a supervisor and PhD student.

Supp mat:

Line 40: "5% of the trails. In"

Line 118: should be "same number of trails per node,"

Fig. S4: Increasing the alpha value for the standard deviation areas would improve the clarity of the graph.

Review form: Reviewer 2

Recommendation

Accept as is

Scientific importance: Is the manuscript an original and important contribution to its field?

Good

General interest: Is the paper of sufficient general interest?

Good

Quality of the paper: Is the overall quality of the paper suitable?

Good

Is the length of the paper justified?

Yes

Should the paper be seen by a specialist statistical reviewer?

No

Do you have any concerns about statistical analyses in this paper? If so, please specify them explicitly in your report.

No

It is a condition of publication that authors make their supporting data, code and materials available - either as supplementary material or hosted in an external repository. Please rate, if applicable, the supporting data on the following criteria.

Is it accessible?

Yes

Is it clear?

Yes

Is it adequate?

Yes

Do you have any ethical concerns with this paper?

No

Comments to the Author

I am satisfied with the new version of the manuscript. I thank the authors for taking into my comments and I also thank them for answering my question thoroughly. I now better appreciate the novelty of the work.

Decision letter (RSPB-2021-0430.R0)

22-Mar-2021

Dear Dr Lecheval

I am pleased to inform you that your manuscript RSPB-2021-0430 entitled "From foraging trails to transport networks: how the quality-distance trade-off shapes network structure" has been accepted for publication in Proceedings B.

The referee(s) have recommended publication, but also suggest some minor revisions to your manuscript. Therefore, I invite you to respond to the referee(s)' comments and revise your manuscript. Because the schedule for publication is very tight, it is a condition of publication that you submit the revised version of your manuscript within 7 days. If you do not think you will be able to meet this date please let us know.

Online supplementary material will also carry the title and description provided during submission, so please ensure these are accurate and informative. Note that the Royal Society will

not edit or typeset supplementary material and it will be hosted as provided. Please ensure that the supplementary material includes the paper details (authors, title, journal name, article DOI). Your article DOI will be 10.1098/rspb.[paper ID in form xxxx.xxxx e.g. 10.1098/rspb.2016.0049].

Sincerely,

Professor Gary Carvalho

Associate Editor

Comments to Author:

Thank you for addressing the comments from myself and the reviewers in such a thorough and comprehensive way. I believe that the manuscript is much improved as a result, and represents an important step forward in the research area.

Reviewer(s)' Comments to Author:

Referee: 1

Comments to the Author(s).

The resubmitted version is a dramatic improvement, and we could suggest only a few minor improvements. We therefore recommend acceptance of this manuscript with only a few adjustments.

The authors addressed the concerns we had on the previous version, providing thorough explanations and modifying the manuscript accordingly.

The new title is more fitting of the story and better highlights the bottom-up approach of the study. All figures have been greatly improved, and so is their new arrangement between the main text and the ESM. The explanation of the K variable makes its effects and behaviour much clearer, in biological terms. The additional replication of the ABM and the presentation of the variance adds trustworthiness to the data without changing the original results.

We particularly like the models with alternative rules and their description (lines 394-410).

Minor points for correction:

Main text:

Line 204: The refence should be to "Electronic Supplementary Materials, Section C" and not to Section A.

Acknowledgements: 3 reviewers! - 'Reviewer 1' is a superorganismic co-reviewer, consisting of a supervisor and PhD student.

Supp mat:

Line 40: "5% of the trails. In"

Line 118: should be "same number of trails per node,"

Fig. S4: Increasing the alpha value for the standard deviation areas would improve the clarity of the graph.

Referee: 2

Comments to the Author(s).

I am satisfied with the new version of the manuscript. I thank the authors for taking into my comments and I also thank them for answering my question thoroughly. I now better appreciate the novelty of the work.

Author's Response to Decision Letter for (RSPB-2021-0430.R0)

See Appendix B.

Decision letter (RSPB-2021-0430.R1)

25-Mar-2021

Dear Dr Lecheval

I am pleased to inform you that your manuscript entitled "From foraging trails to transport networks: how the quality-distance trade-off shapes network structure" has been accepted for publication in Proceedings B.

Your article has been estimated as being 8 pages long. Our Production Office will be able to confirm the exact length at proof stage.

Data Accessibility section

Open Access

Paper charges

Sincerely,

Appendix A

We thank the referees and the editor for giving us the opportunity to improve and resubmit our manuscript. As suggested, we have focused on highlighting the novelty of our work, improving the manuscript organisation and addressing the lack of statistical analysis in our model validation. Our major modifications include (i) a new title and an improved abstract to underscore the novelty of our work, (ii) major changes in the figures chosen to be in the main text and (iii) and additional alternate models specifically developed to improve the reliability of the validation of our model in the justified absence of statistical tests. We hope that these changes successfully highlight the novelty of both our approach, combining models at multiple scales informed and validated by empirical data, and our results, showing how the patterns generated by mechanisms at one scale can explain patterns observed empirically at a larger scale. We detail all these improvements and others in our response below.

Referee: 1

Comments to the Author(s)

Overview:

The authors describe their approach for modelling the generation of multi-source multi-sink networks inspired by the collective behaviour of ants. They find that the well-studied phenomenon of recruitment via positive feedback is pretty much all that is necessary to generate networks that are similar to ant networks. This finding in itself does not sound particularly novel or exciting – but there is much more to this paper than that, and even the authors are risking underselling themselves at some points in the manuscript.

It may seem obvious that positive feedback via pheromone-based recruitment is important for maintaining the internet networks of ants that largely rely on pheromone-based recruitment for foraging. The value of this paper is in taking the rare approach of building up from individual-level first principles to demonstrate how networks are built and maintained at the group-level. This is not just an exercise in model fitting and validation – the authors build their underlying pheromone-recruitment model from a broad range of empirical data, combine this with a network morphogenesis model that itself draws from the parameter space defined by empirical data from wood ants, to produce networks that are validated against the real world networks. It is an elegant demonstration of how emergent group-level properties arise from simple individual-level behaviour. More than that, it is generalisable to more than just the study of wood ant nests. These two key points make the article well worth publishing in Proceedings B, but really should be emphasised in revisions of the manuscript.

The walkthrough of the model design and description of equations was extremely well done – some of the clearest explanations we have seen - and the authors should be commended for their writing, especially here, but also throughout the manuscript.

There are some issues with how the manuscript is arranged, and many missing details for certain aspects of the models, the presentation of results and the significance/discussion of various key features being overlooked. As such, we recommend publication after the listed issues are addressed under a major revision.

Major points:

A major flaw in this manuscript is the arrangement of its presentation – virtually everything of importance is hidden in the ESM, while much less important features are presented in the main text. We have pointed several issues out below, but in general, Figures 1 and 3 should

be ESM, while at least Figs S2, S6 and many points of description in the ESM should be presented in the main text.

We thank the referees for their suggestions regarding the manuscript arrangement. As suggested, we have swapped a few figures between the main text and ESM in order to convey the most important features in the manuscript. Former figures 1 and 3 are now in ESM (Figures S1 and S7) as suggested and we have moved former ESM Figures S7a, S7b, S6c, S6d to the main text (now Figure 3).

Title: You should consider changing the title to better highlight the novelty of your results. As is, it risks underselling your study.

We thank the referees for their comment regarding the title. We changed the former title *Complex biological transport networks are explained by recruitment with positive feedback to* *From foraging trails to transport networks: how the quality-distance trade-off shapes network structure*. We think this new title better highlights the novelty of our result: we show that key structural properties of polydomous transport networks can be explained by the way foraging dynamics address the quality-distance trade-off. It also highlights our novel methodology: it suggests the multiscale analysis (from the individual behaviour to the colony level patterns) performed in our manuscript.

Main text:

Figure 1 appears to serve as an overview of the authors design approach, but this information is already supplied in the text. I would consider removing this figure entirely, or at least exchanging it with one of the more useful figures in the ESM. Also, at it's current position the content in Fig 1 appears long before the reader knows what each of the mentioned studies and models are. The same comment applies to Fig 3 – this Figure doesn't add much value to the manuscript, and could easily be swapped for more informative info in the ESM. So the suggested removal/exchange would add to both clarity and relevance.

We thank the referees for their ideas to improve the manuscript readability and efficiency to convey our messages. We therefore moved former Figures 1 and 3 to the ESM (now Figures S1 and S7).

Lines 153-156: You need to do more to justify the addition of K – as it stands it seems like it could be a useful fudge factor that aids the fitting of your model. It may be entirely appropriate to include “possible persistence or inertia effects” in the model, but why? What references (and ideally, data) support this addition? Prove to me that this isn't just a free parameter to help you fit your model. Or if it is a useful free parameter, explain why it is necessary, and the impact its addition has on the model.

The form of a term for ants leaving the trail $S = a / (K + bX)$ has previously been published in Sumpter & Beekman, 2003 and in Sumpter & Pratt 2003 (reference now added to the main text), although loosely defining the parameter K . In our article we more precisely defined the role of this parameter as accounting “for possible persistence or inertia effects which could be the result of, for instance, the memory of food location over winter or the establishment of physical trails over time improving efficiency and stability of chemical trails.” The presence of

memory of food location over winter and of physical trails in red wood ants has been shown, for instance in (Rosengren, 1971; Gordon, 1992).

We set the value of K to 1 which rapidly becomes much smaller than the term bX as the number of ants on a trail increases — its relative effect in the term $(K + bX)$ thus quickly dissipates and does not help our model to converge to the predictions regarding the distance-quality trade-off discussed in the main text. When there are few foragers, a constant **proportion** $\sim a/K$ of them are lost from the trail. As the number of foragers on the trail increases, this proportion goes down. Finally, when there are lots of foragers, a constant **number** of foragers is lost, $\sim a$. That is, the trail becomes more effective at retaining foragers as traffic increases, but that effectiveness saturates for high traffic levels. We have added this explanation “When there are very few foragers, a constant proportion $\sim a/K$ of them are lost from the trail. As the number of foragers on the trail increases, this proportion decreases. Finally, when there are very many foragers, a constant number of foragers is lost, $\sim a$. That is, the trail becomes more effective at retaining foragers as traffic increases, but that increased effectiveness saturates for high traffic levels.” and both references (Rosengren, 1971; Gordon, 1992) to the paragraph.

Line 169: I'm not convinced that the two models are so similar – you point out some general similarities, but there are some glaring disagreements that aren't discussed. For instance, the biggest disparities appear to be for mediocre food qualities placed at medium distances from the nest (the biggest difference being for large colony sizes at $q=5$, $d=0.5$ and 0.55 , but also see $q=1$, $d=0.4$). The dynamics can appear totally different between the two models, with the ABM appearing to generate stronger trails than the pheromone-recruitment model. The ABM results are averaged over 10 runs, but there is no indication of the variance of these runs. This is especially important for stochastic models with positive feedback, which can generate quite a lot of variance that can run away in the initial stages. For this reason, I am not convinced 10 runs is enough to make an accurate judgement on the average behaviour of the model.

We thank the referees for these two important comments.

1. In our article, we claim that the qualitative predictions of the agent-based model are similar to the ones of the pheromone-recruitment model. In the absence of fine-scale empirical data on the dynamics of trail formation in the presence of varying distance and food quality, we thought it would not be relevant to discuss existing minor qualitative or quantitative differences between our models, since we would not know which one (if any) better depicts reality. We have added a sentence in the ESM section C to highlight differences between both models: “The main difference between the agent-based model and the pheromone-recruitment model is that uncommitted agents in the agent-based model are exploring the foraging area and many of them might be too far from a pheromone trail to sense it, while all uncommitted ants in the pheromone-recruitment model are available to be recruited when the pheromone trail is sufficiently developed”
2. We agree with the referees that the results shown on Figure S4 were not reliable with 10 repetitions and no variance around the mean. We have now performed 100 replications of the 60 conditions shown on this figure and added shaded regions indicating the standard deviation of the mean number of ants on trails. This increase in replication did not result in any qualitative change to our results.

Fig 4 (and others): Much of the data is described as averages (e.g. Fig 4 a, b and c) but there is no indication of the variance. Please add error bars or 95% confidence intervals where necessary. This has been done for 4d.

We thank the referees for their comment which sheds light on the lack of readability of our previous figures and captions. The referees had the impression that Figures 4 a, b, c (now Figures 2 a, b, c) were showing averages, while these figures are actually showing the probability density functions of the variables of interest as estimated by histograms. We enhanced these figures to make it clearer that they are histograms in order to improve their readability and have improved the figure captions.

In 4d, why is there no corresponding empirical data for the 30 and 35 node positions on the x-axis? Perhaps this is an extension of the model beyond the size of the natural networks you could observe? If so, please discuss, especially why it predicts a decrease in average network trail length (perhaps this is an error here, as the red point for 30 nodes has error bars, while none of the other data points for the model do?).

We thank the referees for their observation that in our model, there are a few networks that have more nodes than the maximum observed in nature. We have added the distribution of the total number of nodes per network (Fig. S9b) to show that our model reproduces fairly well the range of the number of nodes found in nature.

There is no statistical analysis of the closeness of the model fit to the empirical observations. While a qualitative comparison is fine, a statistical test of some kind would be instructive (especially in the case of 4c).

We thank the referees for raising an important point regarding the quantification of agreement between empirical observations and model predictions in our manuscript. We would like to answer this by raising several points:

- Our methodology to validate the computational network morphogenesis model against empirical data does not make use of statistical tests, mainly because simulated sample sizes are arbitrarily set (while sample size affects p-value). Very large sample sizes in simulations are known to make p-values meaningless, as even tiny effect sizes result in 'significant' p-values with sufficient replication, and a null hypothesis stating that the system and the model match perfectly is known to be false a priori (White et al., 2014; Hofmann et al., 2018).
- Instead, the methodology we have been following is to compare our models and empirical observations on various emerging metrics. We want to stress here that we are not only comparing those metrics on the basis of their averages but that we are comparing their entire distributions. Therefore, although our comparisons are qualitative in the sense that there are no statistical tests, for a model to perform well against empirical data on many relevant metrics is actually demanding and thus a clear indication of good model performance.

In response to the reviewer's comment about comparison between the model and the empirical data, we acknowledge the need for our analysis to have a baseline of how different

models would perform on these comparisons. Therefore, to better assess whether our model performance can be viewed as 'good', we have thus decided to compare the performance of our network morphogenesis model to two simpler but related models. In our focal model, nodes are connected following a geometric distribution representing a quality-distance trade-off. We created two alternatives to this rule, namely 1. a rule stating that nodes are connected randomly to available nodes; or 2. nodes always connect to the closest available node. Please note that all other aspects of the model remain the same (i.e. same number of nodes, same number of trails per node, same rule forbidding trail intersections). These new control simulations provide additional context for comparisons between our model and empirical observations and we hope they will be helpful to better assess the performance of our model and its biological relevance in the absence of statistical tests.

White, J.W., Rassweiler, A., Samhour, J.F., Stier, A.C. and White, C. (2014), Ecologists should not use statistical significance tests to interpret simulation model results. *Oikos*, 123: 385-388. <https://doi.org/10.1111/j.1600-0706.2013.01073.x>

M. A. Hofmann, S. Meyer-Nieberg and T. Uhlig, "Inferential Statistics and Simulation Generated Samples: A Critical Reflection," *2018 Winter Simulation Conference (WSC)*, Gothenburg, Sweden, 2018, pp. 479-490, doi: 10.1109/WSC.2018.8632306.

ESM:

Line 100: Is the decision not to allow trail intersections based on empirical evidence? If so, please discuss. I feel this is important because network intersections, especially steiner points, are important features of many biological networks;

Latty, T., Ramsch, K., Ito, K., Nakagaki, T., Sumpter, D.J., Middendorf, M. and Beekman, M., 2011. Structure and formation of ant transportation networks. *Journal of The Royal Society Interface*, 8(62), pp.1298-1306.

Liu, L., Song, Y., Zhang, H., Ma, H. and Vasilakos, A.V., 2013. Physarum optimization: A biology-inspired algorithm for the steiner tree problem in networks. *IEEE Transactions on Computers*, 64(3), pp.818-831.

Tero, A., Takagi, S., Saigusa, T., Ito, K., Bebbler, D.P., Fricker, M.D., Yumiki, K., Kobayashi, R. and Nakagaki, T., 2010. Rules for biologically inspired adaptive network design. *Science*, 327(5964), pp.439-442.

If you are "developing a theory for the structure of biological multi-source multi-sink transport networks" (lines 318-319), this could become an important issue.

Indeed, the scarcity of trail intersections is a consistent feature across all colonies mapped in our dataset and also of other polydomous species (Cook, 2014). In the main text, we wrote "There is no trail intersection, in agreement with empirical observations". We added this wording in the ESM as well. This indeed shapes the nature of the networks, which are planar graphs, that is to say that all intersections between edges result in a node (nest or tree). We also add in the conclusion of the discussion: "We note that the networks of the polydomous ant system mapped in the field, rarely showing any trail intersection, are planar graphs without Steiner points (Cook, 2014), unlike networks of polydomous ants studied under laboratory conditions (Latty, 2011) or networks of slime moulds (Tero, 2010). The study of the consequences of the absence of trail intersections and Steiner points in the network structure and properties, for instance by using the modelling framework developed in this article, may help to establish general principles of self-organised biological networks."

Cook, Z., Franks, D.W. & Robinson, E.J.H. Efficiency and robustness of ant colony transportation networks. *Behav Ecol Sociobiol* **68**, 509–517 (2014)

Minor points:

Main text:

Line 49: Can you clarify what you mean by “the scale of the nests?” Do you mean the colony level, or some intermediate level between individual and colony?

This wording is a conclusion following the previous sentence “Workers forage from their habitual nest of origin either to trees (food patches providing hemipteran honeydew), or to other nests of the colony that they treat as food sources”. We mean that the existing literature shows that the absence of a substantial number of workers flowing through the entire network suggests that the processes underlying the self-organised emergence of the network occurs at the level of the nests rather than at the colony level, but without specifying the actual mechanisms (which is the goal of our manuscript). To make the connection between the two sentences, we changed the sentence to “Hence, the resulting multi-source multi-sink transport networks are hypothesised to result from a self-organised process occurring at the scale of the nests”.

Lines 65-67: “Multi-source multi-sink” appears a couple of times within to sentences. Consider rephrasing?

We merged these two sentences into one “Here, we aim to characterise the candidate mechanisms leading to the formation of effective multi-source multi-sink transport networks.”

Line 99: q_i appears in Eq 1, but is not described until Eq 2. Please describe after Eq 1. Also, q_i is described as the quality of the source i , but in Table S1, it is listed as the “Amount of substance”. If amount of substance is directly proportional to the quality of the source i , this needs to be described more clearly.

The sentence introducing Eq 1 has been changed to “In this model (referred to as the Sumpter and Beekman model thereafter), the change in the number of ants X_i committed to a trail i in a situation where the focal nest is connected to J food patches of quality q is”.

Table S1 provides the physical dimensions of the variables used in the pheromone-recruitment model. The unit used for quality in our model is mol, whose dimension is, indeed, an amount of substance. In *Formica lugubris*, primarily feeding on the honeydew of aphids, it is not clear how the qualities of different food patches differ. It is possible that the species of aphids or of trees play a role, for instance in the varying concentration of substances in honeydew — but without more knowledge on these we decided that the number of aphids (presumably with a constant average honeydew production) determines the quality of a food patch — hence represented as an amount of substance. This explanation has been added in ESM, section B.

Line 164: compensate for the

done

Line 164: We were able to find one paper (Fewell et al., 1992. Distance effects on resource profitability and recruitment in the giant tropical ant, *Paraponera clavata*. *Oecologia*) in which authors described recruitment behaviour in giant tropical ants when modifying both distance and quality of food sources.

We thank the referees for this literature suggestion. Indeed, this article describes the recruitment behaviour with varying distance and quality of food sources. However, it does not provide data on “the individual-scale dynamics” as stated in the sentence indicated (L164) because the authors focused on the first recruitment event.

Fig 2: Why no red line for exponential fit for Fig 2b?

We do not predict the distribution of the average quality of food source weighted by the number of recruited ants shown on Figure 2b (now Figure 1b) to follow an exponential distribution, hence the absence of a red line.

Line 278: The explanation for this discrepancy should not be buried in the ESM, but brought here.

We have added a short sentence referring to the ESM paragraph in the main text: “The presence of a peak at $R=0$ in empirical data might be a result of warm and dry weather in recent years (see ESM, section E2).” We left the development of this point in ESM since it is a speculative explanation without firmer ground than an observation.

Line 289: Figure 7c in the ESM is not mentioned in the text. Is this because the model does not seem to match the apparent bimodal distribution of the empirical data? This should be discussed.

We thank the referees for pointing out that a link to figure S7c (now S9) was missing. We added the following sentence in ESM section E Morphogenesis model “This process reproduces fairly well the range of numbers of trees observed in the field (Fig. S9).” To us, this figure is a sanity check that our method to sample stochastically a number of trees per nest is able to recover the range of number of trees per network observed in the field. We do not believe the mismatch with a possibly bimodal distribution to be essential to the results shown in the main text.

ESM:

Line 24: We understand that there is no need for reporting parameter space exploration for each of these. However, it would be nice to have some more information on how these values have been obtained. Are they derived from empirical data (if so, you should provide a reference) or from personal observations?

As pointed out in our manuscript, in the absence of empirical data with a fine scale description of the dynamics of trail formation as a function of quality and distance to food sources, it is difficult to properly calibrate our pheromone-recruitment model. We set the

values of our parameters by choosing relevant orders of magnitude (e.g. we set the time unit to days, in agreement with the time needed for trails to emerge in nature). We checked that the set of parameters gave reasonable values for the time for trail formation to converge to stable trails. In particular, trails converged within 2 days for 73.5% of the trails, within 17 days for 90% of the trails and in more than 50 days for 5% of the trails. We added this information in ESM. Future work will allow for a more thorough exploration of the parameter space.

Line 44: Parameters of the pheromone-recruitment

done

Line 115: have not been accounted for

done

Figure S2: Please provide a description of the red line in the caption.

Done (now fig S3)

Figure S3: The caption should mention that the data you are plotting are averaged over 10 runs (as you explain in Line 42 in the ESM)

Done (now 100 runs)

Figure S5: This figure is particularly difficult to follow. You may find a better way to plot this data (i.e., boxplots)?

Done, thank-you for the suggestion, the former fig S5 has been replaced by boxplots (now figure S6).

Figure S6: This figure may be included in the main text. This way, you can show the distribution of the parameters used in the morphogenesis model without having to refer to the ESM.

We moved figures S6c and d in the main text. Figures S6a and b are now Fig S8a and b.

Figure S7c: The distributions are very different from each other. You should include a note in the caption explaining the reason for such a difference, and/or provide a link to L81-82 in the ESM.

Done (see above)

Referee: 2

Comments to the Author(s)

The manuscript describes a model that implements recruitment with positive feedback as a potential mechanism for transport network morphogenesis in ants. I think that the authors present a potentially useful approach here, but the manuscript requires more work to highlight the novelty of the model introduced here and its potential as a predictive model.

There are several issues the authors need to address :

1) My first concern is about the supplementary material. I found a bit frustrating that essential information and results are only presented in the supplementary material. (on a slighter note, the grammar has not been reviewed carefully throughout the supplementary material - see below for detailed comments)

We thank the referee for the suggestions regarding the supplementary material. As recommended by the referees, we have swapped a few figures between the main text and ESM in order to convey the most important features in the manuscript. Former figures 1 and 3 are now in ESM (Figures S1 and S7) as suggested and we have moved former ESM Figures S7a, S7b, S6c, S6d to the main text (now Figure 3). We also thank the referee for spotting grammar points which have now been corrected.

2) My second concerns is related to the novelty of the work presented here. The authors should better explain how the model proposed in this paper differs from previous models. It was not clear to me what was new in this paper.

Our article tackles a question not really addressed in the existing literature: the behavioural mechanisms underlying the emergence of the polydomous transport networks in ants. To do so, we do use models inspired by previous work but it is the combination of our three models (depicting processes occurring at several scales of description, from the individual ants to the colony level) which is new: to our knowledge, no previous model of polydomous network formation was informed by pheromone-recruitment models. Instead, existing analyses of polydomous network emergence usually use a non-mechanistic framework based on the optimisation of quantities of interests (such as the network efficiency, cost or robustness). Moreover, each of the three models we have used in this study has been modified. The agent-based model incorporates a rather unexplored mechanism (ants without food stochastically returning to nest) with implications for the theoretical distribution of ants around their nest; our pheromone-recruitment model extends a former model by adding the (very important) effect of distance on trail formation. Finally, the network morphogenesis model, although resembling existing ones, is the first one to our knowledge to have mechanistic grounds for its hypotheses. Throughout, we have rooted our models in empirical data, and have been able to validate our model against empirical network properties. We have improved the abstract and changed the title of our manuscript to better highlight the aim and results of our multi-scale modelling to "From foraging trails to transport networks: how the quality-distance trade-off shapes network structure". Additional changes we have made to the text, in particular moving certain key figures to the main text, should also make the novelty and key findings of our study clearer.

3) My third concern regards the ABM presentation. The ABM should be introduced in the manuscript and not only in the SI. In addition, the authors should be more explicit regarding the parameters and assumptions of their ABM (see below for more detailed comments)

We thank the referee for the concern regarding the absence of detailed description of our agent-based model in the main text of our manuscript. The reason for this absence is that the role of this model is to support the core models around which our work is based. It is a model developed specifically to (i) check an hypothesis regarding the distribution of ants around their nest (ii) qualitatively validate the predictions of the pheromone-recruitment model in the absence of empirical data describing the dynamics of trail formation at the scale of the ants with varying distance and quality. Therefore, it has not been calibrated to depict empirical data and its parameters have not been set to fit any empirical data. We hope the changes listed below help make the assumptions of this model explicit. We have also added videos of simulations as supplementary material to help in this regard.

Detailed comments :

Line 90-91 : “the idea that the presence of an ant on a trail can be seen at the colony level as a vote of the individual in favour of the resource connected to the trail.”

=> not entirely sure that this analogy is needed here

We feel this analogy is useful for drawing attention to our focus on the decision-making process, rather than just the spatial distribution of ants.

Line 106-108 : “To examine how the quality of a food source and its distance from the nest interact to influence ant foraging, we reformulate the Sumpter and Beekman model to make all three mechanisms dependent on distance.”

=> should be “To examine how the quality of a food source and its distance from the nest interact to influence ant foraging, we reformulate the Sumpter and Beekman model to make all three mechanisms dependent on distance AND QUALITY”

We thank the referee for this suggestion. However, as described in Eq. 1, the Sumpter and Beekman model is already dependent on quality and the goal of our reformulation of this model is indeed to make the mechanisms dependent on distance, in addition to quality. To make it clearer we specified in the description of Eq 1 that the Sumpter and Beekman model is already encompassing quality with the sentence “In this model (referred to as the Sumpter and Beekman model thereafter), the change in the number of ants X_i committed to a trail i in a situation where the focal nest is connected to J food patches of quality q is”. We also changed the sentence highlighted by the referee to “To examine how the quality of a food source and its distance from the nest interact to influence ant foraging, we reformulate the Sumpter and Beekman model to make all three mechanisms dependent not just on quality, but also distance.”

Line 115 ‘by analysing available published data of single ants *Temnothorax albigipennis* exploring the area around their nest entrance’

=> Why using published data of single ants *Temnothorax albigipennis* exploring the area around their nest entrance, I don't believe this species builds multi-sink multi-source transport networks. Could the authors better justify their choice?

Our aim, by using available data on *Temnothorax albipennis*, is to explore the relevance of our hypothesis that “workers from a given nest looking for food resources (so-called scouts) are not homogeneously distributed in space: they are more likely to be found closer to their nest of origin than elsewhere”. We think that this hypothesis is a general principle in ants. Moreover, the pheromone-recruitment model is not specific to species forming multi-sink multi-source transport networks.

Line 117-119 : Lack of clarity, the sentence should be rewritten.

We thank the referee for giving us the opportunity to reformulate this sentence. We changed the sentence to “A possible underlying behavioural mechanism is that scouts that are still looking for food have a probability of returning to their home nest that remains constant over time (see discussion in electronic supplementary material, section D)”.

Line 127-129: We also assume that recruitment mechanisms mainly rely on pheromone trails, at least in the emergence phase of the trail – other processes possibly being involved after the trail formation, such as workers remembering the location of the food source.
=> something is missing in this sentence.

We changed the sentence to “We also assume that recruitment mechanisms mainly rely on pheromone trails, at least in the emergence phase of the trail; other processes may be involved after the trail formation, such as workers remembering the location of the food source”.

line 138-151: this whole section needs to be re-written to be more explanatory

We hope that the explanations and changes regarding the term for ants leaving the trail detailed to the first referees improved the understandability of this section. These changes are the addition of this explanation “When there are very few foragers, a constant proportion $\sim sd/K$ of them are lost from the trail. As the number of foragers on the trail increases, this proportion decreases. Finally, when there are very many foragers, a constant number of foragers is lost, $\sim sd$. That is, the trail becomes more effective at retaining foragers as traffic increases, but that increased effectiveness saturates for high traffic levels.” and of these two references (Rosengren, 1971; Gordon, 1992) to the paragraph.

Line 159: the authors should better introduced the weighting parameters. What is the sensitivity of these parameters ? How was the model calibrated ?

As pointed out in our manuscript, in the absence of empirical data with a fine scale description of the dynamics of trail formation as a function of quality and distance to food sources, it is difficult to properly calibrate our pheromone-recruitment model. We set the values of our parameters by choosing relevant orders of magnitude (e.g. we set the time unit to days, in agreement with the time needed for trails to emerge in nature). We checked that the set of parameters gave reasonable values for the time for trail formation to converge to stable trails. In particular, trails converged within 2 days for 73.5% of the trails, within 17 days for 90% of the trails and in more than 50 days for 5% of the trails. We have added this information in ESM. We did not run a sensitivity analysis but show predictions of simulations

of the pheromone-recruitment for another set of parameters depicting ants less able to efficiently lay and follow pheromone trails. Future work will allow for a more thorough exploration of the parameter space.

The authors never refers to figure S7c. (or I missed it !)

We thank the referee for pointing out that a link to figure S7c (now S9) was missing. We added the following sentence in ESM section E Morphogenesis model “This process reproduces fairly well the range of numbers of trees observed in the field (Fig. S9).” To us, this figure is a sanity check that our method to sample stochastically a number of trees per nest is able to recover the range of number of trees per network observed in the field.

Figure 3: I don't see how this figure can help the reader. It is by far more interesting to have Fig S6 and S7 in the manuscript instead of this figure. Figure 3 could be moved in the SI.

We thank the referee for his suggestion regarding the manuscript organisation. We moved former Figures 1 and 3 in ESM (now respectively Figs S1 and S7) and Figs S6c,d and S7a,b to the main text (now respectively Fig 4 c, d, a, b).

Line 269-291 : all comparisons are qualitative, is there a way to quantify if the model properly fit the data.

We thank the referee for raising an important point regarding the quantification of agreement between empirical observations and model predictions in our manuscript. We would like to answer this by raising several points:

- Our methodology to validate the computational network morphogenesis model against empirical data does not make use of statistical tests, mainly because simulated sample sizes are arbitrarily set (while sample size affects p-value). Very large sample sizes in simulations are known to make p-values meaningless, as even tiny effect sizes result in 'significant' p-values with sufficient replication, and a null hypothesis stating that the system and the model match perfectly is known to be false a priori (White et al., 2014; Hofmann et al., 2018).
- Instead, the methodology we have been following is to compare our models and empirical observations on various emerging metrics. We want to stress here that we are not only comparing those metrics on the basis of their averages but that we are comparing their entire distributions. Therefore, although our comparisons are qualitative in the sense that there are no statistical tests, for a model to perform well against empirical data on many relevant metrics is actually demanding and thus a clear indication of good model performance.

In response to the reviewer's comment about comparison between the model and the empirical data, we acknowledge the need for our analysis to have a baseline of how different models would perform on these comparisons. Therefore, to better assess whether our model performance can be viewed as 'good', we have thus decided to compare the performance of our network morphogenesis model to two simpler but related models. In our focal model, nodes are connected following a geometric distribution representing a quality-distance trade-off. We created two alternatives to this rule, namely 1. a rule stating that nodes are connected randomly to available nodes; or 2. nodes always connect to the closest available

node. Please note that all other aspects of the model remain the same (i.e. same number of nodes, same number of trails per node, same rule forbidding trail intersections). These new control simulations provide additional context for comparisons between our model and empirical observations and we hope they will be helpful to better assess the performance of our model and its biological relevance in the absence of statistical tests.

White, J.W., Rassweiler, A., Samhouri, J.F., Stier, A.C. and White, C. (2014), Ecologists should not use statistical significance tests to interpret simulation model results. *Oikos*, 123: 385-388. <https://doi.org/10.1111/j.1600-0706.2013.01073.x>

M. A. Hofmann, S. Meyer-Nieberg and T. Uhlig, "Inferential Statistics and Simulation Generated Samples: A Critical Reflection," *2018 Winter Simulation Conference (WSC)*, Gothenburg, Sweden, 2018, pp. 479-490, doi: 10.1109/WSC.2018.8632306.

Line 326: "While the topology and the functions of the resulting networks differ between the ant, fungal and slime mould systems"

=> The function of fungal and slime mould networks is also to transport resources, no? The topology is indeed different, but the function remains the same.

We have changed the sentence to "While the topology of the resulting networks differ between the ant, fungal and slime mould systems, the striking resemblance of their underlying mechanisms suggest a unifying theory of self-organised biological networks based on the combination of exploration and positive reinforcement of best sources."

Supplementary material :

Line 16-17: "When a food source is found, the successful ant returns to the nest and deposits a quantity of pheromones which is proportional to the quality of the food source."

=> what path the ant is following when it returns to the nest?

In this model, agents are returning to the nest following a straight line between the source and the nest. We improved the sentence, now written as "When a food source is found, the successful ant returns to the nest following a straight line between the source and the nest and deposits a quantity of pheromones which is proportional to the quality of the food source."

Line 19-20 : When these cells have pheromones, ants follow the gradient of pheromone concentration by stochastically choosing the cell with the highest concentration with probability $p = 0.95$.

=> this probability seems quite high, could you justify such a high value empirically

As mentioned in the main text, we do not have empirical data to calibrate the parameters of the agent-based model. We set this value in order to put some noise in our simulations. Changing this value would only change the levels of noise in our simulations.

Line 20-21 : “At each time step without pheromone in the three cells ants face, they have a set probability, constant over time, to return to the nest.”

=> what is the value of this probability

We thank the referee for spotting the absence of this probability. We changed the sentence to “At each time step without pheromone in the three cells ants face, they have a set probability equal to 0.90, constant over time, to return to the nest.” We have added video as supplementary material to show more explicitly the effect of this parameter on the distribution of ants around their nest.

Fig S1 : “Distribution of the rank of distance of the source selected by the colony, i.e. the source with the most crowded trail (bars), and fitted theoretical geometric distribution (dots) in simulations of the pheromone-recruitment model (parameters in Methods).”

=> please rephrase for clarity

We thank the referee for spotting this unclear legend for figure S1 (now figure S2). We changed it to Distribution of the rank of distance of the exploited food source, i.e. the source with the most crowded trail (bars), and fitted theoretical geometric distribution (dots) in simulations of the pheromone-recruitment model (parameters in Methods). Rank of distance is 1 for the closest food source available, 2 for the second closest and so on.

Fig S2 : “for ants less efficient in pheromone communication “

=> could you be more explicit here.

The sentence following the one spotted by the referee gives more information about how we set parameters to represent ants with less efficient abilities in pheromone communication: “Namely, this set of parameters simulates ants less effective in following a pheromone trail and with reduced pheromone range”, along with values of parameters.

Fig S2: the legend is illegible for the first panels, the authors should move it to the bottom right corner

done

Here are some grammar errors I spotted in the SI :

Fig S6 : “Number of nest per colony” (x)-axis

=> should be “Number of nestS per colony”

done

Fig S6 : “(d) respectively show the distributions of the number of nest per colony”

=> should be “number of nestS per colony”

done

Fig 7: “Grey dotted line show the distribution used as input in the model.”

=> should be “Grey dotted line showS the distribution used as input in the model”

done

Fig 7b: "Number of internest trail per nest" (x)-axis
=> should be "Number of internest trailS per nest"

done

Fig 7c: "Number of tree" (x)-axis
=> should be "Number of treeS"

done

Appendix B

We thank the editor and the referees for their important contributions in improving our article. All minor points for correction raised by Referee #1 have been addressed.